# Evaluation of In Vitro Protein Hydrolysis in Seven Insects Approved by the EU for Use as a Protein Alternative in Aquaculture

**DOI:** 10.3390/ani14010096

**Published:** 2023-12-27

**Authors:** María Rodríguez-Rodríguez, María José Sánchez-Muros, María del Carmen Vargas-García, Agnes Timea Varga, Dmitri Fabrikov, Fernando G. Barroso

**Affiliations:** 1Department of Biology and Geology, CECOUAL, University of Almería, Carretera de Sacramento s/n, 04120 Almeria, Spain; fbarroso@ual.es; 2Department of Biology and Geology, CEIMAR, University of Almería, Carretera de Sacramento s/n, 04120 Almeria, Spain; mjmuros@ual.es (M.J.S.-M.); mcvargas@ual.es (M.d.C.V.-G.); avarga@ual.es (A.T.V.); df091@ual.es (D.F.)

**Keywords:** insect meal, digestibility, hydrolysis degree, *Tenebrio molitor*, *Hermetia illucens*

## Abstract

**Simple Summary:**

In aquaculture, fishmeal and soybean meal have traditionally been used in the formulation of feed to cover the protein demands of fish. More sustainable protein sources, such as algae and insects, are currently being investigated as alternatives. Therefore, this study evaluates the in vitro digestibility of protein from seven insects approved by the European Union for use in animal feed. The study results suggest that *Tenebrio molitor* had digestibility similar to that of fishmeal, while *Acheta domestica* and *Hermetia illucens* obtained similar data to those provided by soybean meal.

**Abstract:**

Rapid population growth is leading to an increase in the demand for high-quality protein such as fish, which has led to a large increase in aquaculture. However, fish feed is dependent on fishmeal. It is necessary to explore more sustainable protein alternatives that can meet the needs of fish. Insects, due to their high protein content and good amino acid profiles, could be a successful alternative to fishmeal and soybean meal traditionally used in sectors such as aquaculture. In this work, seven species of insects (*Hermetia illucens*, *Tenebrio molitor*, *Acheta domestica*, *Alphitobius diaperinus*, *Gryllodes sigillatus*, *Gryllus assimilis*, and *Musca domestica*) approved by the European Union (UE) for use as feed for farmed animals (aquaculture, poultry, and pigs) were studied. Their proximate composition, hydrolysis of organic matter (OMd), hydrolysis of crude protein (CPd), degree of hydrolysis (DH/NH_2_ and DH/100 g DM), and total hydrolysis (TH) were analyzed. The results showed that *Tenebrio molitor* had digestibility similar to that of fishmeal, while *Acheta domestica* and *Hermetia illucens* provided similar digestibility to that of soybean meal. The acid detergent fiber (ADF) data were negatively correlated with all protein digestibility variables. The differences in the degree of hydrolysis (DH) results and the similarity in total hydrolysis (TH) results could indicate the slowing effects of ADF on protein digestibility. Further in vivo studies are needed.

## 1. Introduction

Seven insect species, *Hermetia illucens* (Linneo, 1758) (black soldier fly), *Tenebrio molitor* (Linneo, 1758) (yellow mealworm), *Acheta domestica* (Linneo, 1758) (house cricket), *Alphitobius diaperinus* (Panzer, 1797) (lesser mealworm), *Gryllodes sigillatus* (Walker, 1869) (banded cricket), *Gryllus assimilis* (Fabricius, 1775) (two-colored cricket), and *Musca domestica* (Linneo, 1758) (housefly), are allowed to be used in the production of feed for farmed animals (aquaculture, poultry, and pigs) by Commission Regulation (EU) 2017/893 of 24 May 2017 [1]. In November 2021 (EU Regulation 2021/1925), *Bombyx mori* was added as the eighth species approved for this purpose [2].

The rapid growth of the population makes it necessary to look for protein alternatives that can meet our needs without compromising the environment. Aquaculture provides approximately 50% of fish production and is expected to increase in the coming years. Insects appear to be an appropriate replacement for fishmeal and soybean meal traditionally used in aquaculture [3,4,5,6] due to their high crude protein content (40–70% dry weight) that meets the WHO (World Health Organization) requirements for essential amino acids, minerals, and vitamins, in addition to their proportion of polyunsaturated and saturated fatty acids [7].

Insects have better conversion efficiency (transforming the food they eat into their own body weight) than livestock or fish [8,9]. They reproduce easily, and because they can be reared in smaller areas (1 hectare for *T. molitor* compared to 3.3 hectares on average for pigs or chickens and 10 hectares for cattle to produce the same amount of protein), they also have a higher yield per hectare than other crops. Insects also produce fewer greenhouse gas and ammonia emissions per kg of meat than pigs or cattle (1 kg of beef emits 14.8 kg of CO_2_; chickens and pigs emit 1.1 kg and 3.8 kg, respectively; and *T. molitor* larvae, locusts, and crickets emit 100 times fewer greenhouse gases and 10 times less ammonia) [8,10,11]. Insects also have other advantages related to their economic potential, such as short breeding cycles and high reproduction rates [12].

When studying insects as food, one must consider not only their amino acid composition but also their digestibility [13]. Digestibility requires in vivo experiments; however, current animal welfare regulations encourage a reduction in the use of animals for experimental purposes and the replacement of animals with alternative techniques. In vitro digestibility experiments make it possible to reduce the number of animals to be used for testing each source. In addition, in vitro experiments can be carried out to simulate enzyme, pH, time, and temperature conditions. The information obtained can serve as a starting point for in vivo experiments. In addition, in vitro protein hydrolysis makes it possible to compare the degradation of proteins from traditionally eaten foods, such as soy or fish, with insects. When discussing digestibility, special attention should be paid to chitin, a polysaccharide that forms part of the exoskeletons of insects. Chitin is a polymer of N-acetyl-glucosamine joined by a β(1-4) glycosidic bond, which is a crude fiber and, therefore, not digestible by monogastric animals [14], although the chitinase enzyme is found in gastric juices [10]. Chitin can negatively interfere with nutrient absorption [15], specifically protein utilization [14]. The composition of chitin depends on the order, species, and stage of the life cycle. In the larval stage, insects have relatively soft structures with less outer layer content formed by sclerotized proteins and chitin, making them more digestible [16]. Low protein and lipid digestibility related to chitin content have been studied in fish [17]. Furthermore, the low in vitro digestibility reported in the literature is related to the presence of highly sclerotized or chitin-bound amino acids that make such acids difficult to digest [18]. Under this background, as suggested by Gómez et al. [12], insect meal does not have to replace 100% of the usual dietary ingredients with insects; instead, insects can be introduced as a component of the meal. Much research has been performed in aquaculture on the replacement of protein in diets with insect meal to find a ratio that is not detrimental to health and growth.

This research aims to compare the digestibility of the in vitro protein of seven insect species approved by the EU for use in aquaculture. The results may provide further criteria for selecting which insect species might be most suitable as an alternative to fishmeal or soybean meal.

## 2. Materials and Methods

Nymphs from orthopterans *A. domestica*, *G. sigillatus*, and *G. assimilis*, larvae from the coleopterans *T. molitor* and *A. diaperinus*, and larvae and pupae of dipterans *H. illucens* and *M. domestica* were used. Insects were purchased from Entomotech S.L. (Almeria, Spain) (*H. illucens*), Bioflytech S.L. (Murcia, Spain) (MD), and Grupo Insectem S.L. (Valencia, Spain) (*T. molitor*, *A. diaperinus*, *A. domestica*, *G. sigillatus*, and *G. assimilis*) and killed by freezing (−18 °C) (Infiniton FG-246W, Infiniton, Granada, Spain). The nymphs were oven dried at 100 °C (Selecta Dryterm 2000787, JP Selecta, Barcelona, Spain) to a constant weight and degreased with pure diethyl ether (309966 Sigma-Aldrich, St. Louis, MO, USA). Subsequently, the nymphs were ground at the Technical Services of the University of Almeria (Spain) with a particle size between 0.025 and 0.08 mm. Fishmeal and soybean meal were used as protein references in the aquaculture feed and were provided by the Experimental Diet Service of the University of Almeria.

### 2.1. Proximate Composition

Analyses were performed according to the Association of Official Analytical Chemists [19]. Dry matter and ash were determined via gravimetry after drying at 105 °C (24 h) (#934.01) in a conventional oven and in a combustion oven at 500 °C (6 h) in a muffle furnace (Selecta R-3L) (#942.05). The protein and ether extract contents were determined via the Kjeldahl method (Selecta Pro-Nitro S 4002851) (Nx4.79 for insects and Nx6.25 for fishmeal and soybean meal) (#954.01) and the Soxhlet technique (Selecta fat extractor 4002842) (#920.39). Acid detergent fibers were determined using the Ankom technique (Ankom200) [20], which consists of digestion with H_2_SO_4_ and cetyl trimethylammonium bromide (CTAB). The fiber residues were predominantly composed of cellulose and lignin. All analyses were performed in triplicate.

### 2.2. In Vitro Protein Hydrolysis

The methodology described by Ottoboni et al. [21] was followed with some modifications. The weight corresponding to 600 mg of protein samples was introduced with 80 mL of 0.075 M HCl. The mixture was incubated at 37 °C for 15 min in a shaking water bath. The gastric phase was carried out using 2 mg/mL porcine pepsin (P7000 Sigma-Aldrich, St. Louis, MO, USA) for 240 min under constant agitation at 37 °C. After gastric digestion, the pH was adjusted to 7.5 with a 0.2 M NaOH solution, and the intestinal phase was carried out by adding 80 mL of the phosphate buffer (pH 7.5, 0.2 M) with 1.5 mg/mL of porcine pancreatin (P1750 Sigma-Aldrich) for 240 min under constant agitation at 37 °C.

Before adding enzymes from the gastric and intestinal phases, samples were taken to be used as blanks. In this way, the data from both phases were not affected by the amino acids present at time 0. As the intestinal phase occurred after the gastric phase, intestinal digestibility was taken to include the sum of the gastric and intestinal phases.

The reaction of the collected samples was stopped with an equal volume of 20% trichloroacetic acid (TCA) every 30 min throughout the gastric and intestinal phases. Three replicates per sample were analyzed.

### 2.3. Organic Matter Digestibility (OMd)

After in vitro hydrolysis, the samples were centrifuged. The samples were washed with 80 mL of distilled water and centrifuged again. The residue was dried at 100 °C for 24 h to calculate the dry matter and then at 500 °C to obtain the ash of the residue. The following formula was used to calculate the digestibility of the organic matter:(1)$$OM\;Digestibility(\%)=\frac{{(DM}_{i}-{Ash}_{i})-{(DM}_{f}-{Ash}_{f})}{{(DM}_{i}-{Ash}_{i})}\times 100$$
where DM_i_ is the initial dry matter; DM_f_ is the final dry matter expressed in grams; Ash_i_ is the initial ash of the samples; and Ash_f_ is the final ash of the samples after hydrolysis.

### 2.4. Crude Protein Digestibility (CPd)

Dry residue corresponding to the digestibility of the matter was subjected to the Kjeldahl method to calculate crude protein after in vitro hydrolysis. The following formula was used to calculate the digestibility of the crude protein:(2)$$CP\;Digestibility(\%)=\frac{{CP}_{i}-{CP}_{f}}{{CP}_{i}}\times 100$$
where CP_i_ is the initial crude protein and CP_f_ is the crude protein after hydrolysis, expressed in grams.

### 2.5. O-Phthaldialdehyde (OPA) Method

The OPA methodology described by Church et al. [22] was used to calculate the α-amino groups using L-leucine as a standard. α-amino groups are found in most amino acids that comprise proteins. This method measures the α-amino groups released after proteolysis.

The OPA results are expressed as the percentage of free amino groups in relation to the total amino groups (DH/NH_2_) and the percentage of free amino groups in relation to dry matter (DH/DM) according to the following calculation.

The degree of hydrolysis (DH/NH_2_) relates the number of peptide bonds broken during hydrolysis (h) to the total number of peptide bonds present in the sample (h_tot_).
(3)$$DH/{NH}_{2}(\%)=\frac{h}{htot}\times 100$$
where h was calculated by measuring the free amino groups via OPA at the end of hydrolysis (480 min), and h_tot_ is the total number of amino groups in the sample [23]. These quantities were determined by hydrolyzing 50 mg of the initial sample with 2.5 mL of 6 M HCl for 24 h at 100 °C.

DH/100 g of DM refers to the number of peptide bonds broken during protein hydrolysis (h) per 100 g of dry matter.
(4)$$DH/DM\;(\%)=\frac{\left[h\right]}{{[DM}_{i}]}\times 100$$
where [h] was calculated by measuring the free amino groups with OPA at the beginning of hydrolysis (time 0) and after the end of the gastric (240 min) and intestinal (480 min) phases, expressed per milliliter, and [DM_i_] is the organic matter introduced at the start of protein hydrolysis per milliliter.

Time 0 was used as a blank for the gastric and intestinal phases, while the intestinal phase here is the sum of the gastric and intestinal phases.

The results are expressed as dry matter to measure the yield of the sample since the purpose of these meals is to be used in feed as a substitute for other proteins.

### 2.6. Total Hydrolysis (TH)

The degree of hydrolysis only measures the broken peptide bonds at the end of in vitro hydrolysis (480 min). However, the small peptides present that are susceptible to the end of hydrolysis were not considered. Therefore, to calculate total hydrolysis, the final hydrolysis supernatant was completely hydrolyzed with the same amount of 12 M HCl for 24 h at 100 °C and neutralized with 6 M NaOH.
(5)$$Total\;hydrolysis\;(\%)=\frac{supernatant\;completely\;hidrolyzed}{htot}\times 100$$

### 2.7. Statistical Analysis

The experimental results were expressed as the means ± SD. Statistical differences in insect digestibility were analyzed using an ANOVA one-way test, followed by a comparison of means (Tukey test). The correlations were analyzed using multivariate pairwise correlation analysis (IBM SPSS Statistics v29.0.1.0.).

## 3. Results

### 3.1. Proximate Composition

Proximal composition data are reported in Table 1. Insects have a high crude protein content. All species had higher values than those of soybean meal. Furthermore, the results for *G. sigillatus*, *A. domestica*, *A. diaperinus*, and *T. molitor* were higher than those of fishmeal. *H. illucens* had the highest ether extract content, and all insects yielded higher results than those obtained for fishmeal and soybean meal. ADF was similar in all meals but was higher in *M. domestica*. Fishmeal had the highest ash content. A higher ash content was also recorded in *H. illucens* compared to that in insects and soybean meal.

### 3.2. Organic Matter Digestibility (OMd)

OMd data (Figure 1) exceeded 90%, except in *M. domestica*. Fishmeal had the highest value, while soybean meal was higher than only *M. domestica*. Insects with the highest data were *T. molitor*, *G. sigillatus*, and *A. diaperinus* (≥94%) and provided similar results to those of fishmeal.

### 3.3. Crude Protein Digestibility (CPd)

CPd data (Figure 2) exceeded 90%, except for *M. domestica*. Soybean meal had the highest results, while *M. domestica* had the lowest. All other insects had results similar to those of fishmeal.

### 3.4. Degree of Hydrolysis (DH/NH_2_) and Total Hydrolysis (TH)

The DH data showed large differences between the samples. No significant differences in DH were observed between the fishmeal, soybean meal, and *T. molitor* (Figure 3). *T. molitor* larvae offered the highest results, followed by *H. illucens* and *A. domestica*. The lowest results were obtained for *G. sigillatus*, *M. domestica*, and *A. diaperinus*. TH values were higher in samples with lower DH values (Figure 3). Consequently, the total digestibility was the same for *H. illucens*, *G. assimilis*, *A. domestica*, and *T. molitor*, with similar results to those of fishmeal. All insects exceeded the soybean meal data, except for *A. diaperinus*, which had the lowest TH value.

### 3.5. Gastric and Intestinal Digestion

Figure 4 shows the screening results of the amino groups released per 100 g of dry matter during in vitro protein hydrolysis. In both phases of hydrolysis, the greatest release of amino groups occurred during the first 30 min and tended to stabilize at about 120 min. As shown in Table 2, at the end point, *T. molitor* had similar results to those of fishmeal. Furthermore, *A. domestica* and *H. illucens* had higher results than those of soybean meal. *G. sigillatus*, *A. diaperinus*, and *M. domestica* had the lowest yields.

### 3.6. Correlations

Table 3 shows weak correlations between OMd and CPd (r = 0.4509). The negative correlation between ADF and all digestibility indices is particularly notable (DH, r = −0.5981; OMd, r = −0.6456; CPd, r = −0.5970), except for TH (r = nsd). The highest correlation was obtained between the protein digestibility variables DH and TH (r = 0.7016).

## 4. Discussion

Clearly, insects will play a key role in the feeding of aquacultured fish in the coming years. Although the production of insects is more sustainable and, in general, insects have higher protein contents and amino acid profiles than other alternative feeds to fishmeal, there are large gaps in general knowledge that prevent insects from being fully exploited. In this sense, it is essential to determine the extent to which the protein content of these insects is digestible by fish. In vivo tests are ideal, but concern for the welfare of fish makes it necessary, at least in preliminary studies, to use in vitro techniques to gain initial knowledge without having to use a significant number of fish.

The potential of insects as a protein food source was validated by our results, as many of the insect species analyzed offer a high percentage of protein (55.4–72.0%) (Table 1), similar to or greater than that of fishmeal and soybean meal. However, one drawback is the high content of ether extract. Barroso et al. [24] noted that insects offer higher EE data compared to those of fishmeal and soybean meal. EE is influenced by stage, season, sex, environment, insect diet, and post-harvest methods. According to Mastoraki et al. [25], insect larvae contain about 1–10% ADF. In our study, all species contained less than 10%, and no differences were found between larvae and adults. *M. domestica* had the highest ADF value, as expected, in the pupa stage. Kovitvadhi et al. [26] also obtained higher ADF values for *M. domestica* than for *H. illucens*, *T. molitor*, *A. domestica*, or fishmeal. The high ash content of fishmeal may be due to the presence of bones or scales. In general, insects have a similar proximal composition to fishmeal and soybean meal in terms of protein and ADF levels. However, fish have a higher EE content, which can be easily reduced by defatting insect meals.

### 4.1. Organic Matter Digestibility (OMd) and Crude Protein Digestibility (CPd)

The insects studied presented highly digestible organic matter (Figure 1). In other reported data simulating in vitro digestibility in ducks, insects had higher OMd data than those of the reference proteins (fishmeal and soybean meal) [26,27]. Poelaert et al. [28] used commercial porcine enzymes and obtained higher data for insects than for soybean meal. According to Bosch et al. [29], fishmeal presented higher OMd data than those of soybean meal. *M. domestica* also had values below those of *H. illucens*, *A. domestica*, and *T. molitor*. No further studies on OMd in insect meal were found in the literature.

The digestibility data obtained via the Kjeldahl method in the literature vary according to the in vitro digestibility method followed: for fishmeal, 68–84.9% [25,29]; for soybean meal, 60.8–94.7% [26,29,30]; for *A. diaperinus*, 91.5% [30]; for *A. domestica*, 57–91.7% [26,29,30,31]; for *H. illucens*, 67–89.7% [29,30,32]; and for *T. molitor*, 60–92.5% [26,28,29,30,32,33,34,35]. In general, high percentages of around 90% were found (Figure 2), similar to the data obtained in this investigation. Furthermore, similar to the data obtained by Bosch et al. [29], soybean meal performed better than fishmeal.

Insects presented values similar to those of the reference proteins, except for *M. domestica*, which had lower matter and protein digestibility. However, there seems to be no difference in insect order. The stage of the life cycle also does not appear to be affected. Bosch et al. [29] obtained better results for *H. illucens* larvae than for *H. illucens* pupae in both OMd and CPd. In contrast, Kovitvadhi et al. [26], studying digestibility in ducks, obtained better results for OMd and CPd in *Bombyx mori* pupae compared to the results in larvae.

### 4.2. Degree of Hydrolysis (DH) and Total Hydrolysis (TH)

DH data in the literature vary depending on the in vitro hydrolysis methodology used. In some cases, insects outperformed reference proteins, such as in Manditsera et al. [15], in which whey protein corresponded to a percentage of 34% while *Locusta migratoria* was 36.4%; other insects, such as adults of *Eulepida mashona* and *Henicus whellani*, had lower percentages (30.6% and 29.7%, respectively). Also, the data provided by Lee et al. [36] showed higher DH values in *Protaetia brevitarsis* larvae than those in beef meat (54.9% and 50.6%, respectively).

In this case, *T. molitor* has values similar to those of fishmeal and soybean meal, while *G. sigillatus*, *A. diaperinus*, and *M. domestica* clearly have the lowest values (Figure 3). Conversely, in other studies, *T. molitor* was outperformed by other insects. In Janssen et al. [37], *H. illucens* and *A. diaperinus* outperformed *T. molitor* (22%, 15.8%, and 14.9%, respectively). In Zielińska et al. [38], *G. sigillatus* adults and *Schistocerca gregaria* outperformed *T. molitor* larvae (32%, 30.5%, and 14.8%, respectively). These differences could be due to several factors, such as the quality of the reagents or the different methodologies used, as well as the use of trypsin in the intestinal phase in Janssen et al. [37] instead of pancreatin (which contains trypsin, amylase, lipase, ribonuclease, and proteases), which was used in the present study, among other factors.

Remarkably, phylogenetically similar species such as *T. molitor* and *A. diaperinus* (family Tenebrionidae) and *A. domestica* and *G. sigillatus* (family Gryllidae) showed different DHs, although we expected more similar results. These differences seem to occur between species, regardless of the order in which they belong.

The similarities between the TH results indicate that even insects with low DH values (few amino groups released) ultimately digested the same amount of protein. The data in Figure 4 show that soybean meal could have been digested more efficiently, with almost equal DH and TH results. The same factor was observed for fishmeal and *T. molitor*, with a difference of approximately 3% between the two variables.

*T. molitor* had a TH value similar to that of fishmeal. Similar values were also obtained for *H. illucens*, *G. assimilis*, *A. domestica*, and soybean meal. Finally, *A. diaperinus* obtained the lowest data.

In general, *T. molitor* appears to show the most similar protein hydrolysis to fishmeal. This result is in agreement with several in vivo feeding experiments carried out on sea bream (*Sparus aurata*) [39] and trout (*Oncorhynchus mykiss*) [40], where fishmeal was replaced by 50% *T. molitor* or *H. illucens*. In trout, Melenchón et al. [40] did not obtain significant differences in production indices between fish fed the control diet and those fed a diet with 50% *T. molitor*. In sea bream [39], although the indices of fish fed a *T. molitor* feed (50%) were slightly lower than those of the control fish, they were much more similar than those of fish fed an *H. illucens* substitution (50%).

The differences in digestibility (DH) among the insects studied may be due to ADF. As indicated above, in the present analysis, a negative correlation was found between ADF and the different digestibility variables studied (OMd, CPd, and DH) (Table 3), but there was no such correlation with the TH data. Chitin may reduce the efficiency of digestive enzymes on the substrate [40]. The results could indicate that ADF slows the release of amino groups (digestibility) but does not prevent protein digestibility.

Differences found between DH and TH (Figure 4) could indicate that an insufficient in vitro hydrolysis time was reached, and, therefore, not all amino groups of the solubilized protein were broken down. However, as shown in Figure 3, the amino groups that presented a constant trend in screening are striking. Purschke et al. [41] used alcalase, neutrase, and flavorzyme as enzymes, finding a constant trend between minutes 300 and 450. However, as hydrolysis continued, a considerable increase in digestibility was found at minute 1500. Despite this result, the inclusion of fish hydrolysates in fish and crustacean diets was observed to yield a higher absorption of dipeptides and tripeptides than free amino acids [42]. However, this absorption also increases compared to that in diets without hydrolysates, in addition to an increase in growth and feeding efficiency [43]. On the other hand, low-molecular-weight peptides present immune-stimulating and antibacterial properties that primarily stimulate nonspecific immune responses [44].

Experimental designs in vivo are needed to elucidate whether DH underestimates final protein hydrolysis. Such analyses would indicate whether the breakdown of amino groups does or does not have the same mechanics in all species, with different processes underlying the release of amino acids, dipeptides, tripeptides, etc.

### 4.3. Gastric and Intestinal Digestion

The amino groups were quantified at time 0 before enzymatic action (Table 2). Other studies also reported some solubilization at the beginning of protein hydrolysis, with 20% for *G. sigillatus* [45] and 43% for *H. illucens* [46]. The authors noted that the use of different buffers and pH values can influence the initial amount of amino groups released (without enzymatic action). A possible cause of this solubilization could be autolysis. Grinding releases endogenous enzymes from insects [37], which can lead to the degradation of insect muscle and exoskeleton proteins [47]. Studying autolysis in the bee body, Bishop et al. [48] observed increased autolysis under an acidic pH. This result may be due to the activation of autolytic enzymes under an acidic pH [49]. In our case, at the beginning of protein hydrolysis, we solubilized the insect meal in an HCl solution before the addition of pepsin. This factor could explain the high values for free amino groups observed at the beginning of hydrolysis in some insects (Table 3). Notably, insects with a higher amount of amino groups before starting protein hydrolysis also presented better digestibility data at the end of hydrolysis, as in the case of *T. molitor*, *H. illucens*, *G. assimilis*, and *A. domestica*, compared to the low initial and final data presented by *A. diaperinus*, *M. domestica*, and *G. sigillatus*. This result could indicate that the degradation mechanisms occurring inside insects could help their subsequent digestibility.

Barroso et al. [24] studied the amino acid profiles of various insect species corresponding to different orders, finding that the proportion of amino acids was not only a specific characteristic but also conditioned by the taxonomic group to which the insect belonged. In the generated dendrogram, Diptera (with seven species analyzed, including pupae and larvae of *H. illucens* and *M. domestica*) were grouped and showed the most similar amino acid profiles to those of fishmeal, while Coleoptera (four species including *T. molitor*) and Orthoptera (five species including *G. assimilis* and *A. domestica*) were grouped at a greater distance. Therefore, we would expect the release of amino groups to be similar among species of the same order. However, it appears that the release of amino groups is more strongly dependent on specific characteristics, as there are notable differences between species belonging to the same family. Thus, while *T. molitor* showed the highest release of amino groups in both the gastric and intestinal phases, that of *A. diaperinus*, a species also belonging to the Tenebrionidae family, showed the lowest release (Figure 4). It should also be noted that the species currently most commonly used in animal feed (*T. molitor*, *H. illucens*, and *A. domestica*) are also the species with the highest amino groups released, even higher than those in soybean meal.

In a study comparing the digestibility of the gastric and intestinal phases of *T. molitor*, during the first 30 min, maximum digestibility was reached in the gastric phase, while, in the intestinal phase, maximum digestibility was reached between 60 and 120 min [47]. No other studies on insects were found.

*T. molitor*, *A. domestica*, and *H. illucens* presented similar amino groups released in dry matter, such as fishmeal and soybean meal. Consequently, these species would need the same amount of feed to obtain similar digestibility results.

### 4.4. Correlations

According to Marono et al. [32], high ADF values can translate into lower digestibility due to their negative correlation. This fact was corroborated in our study since ADF data were also negatively correlated with all digestibility variables studied (Table 3). Therefore, when it is not possible to determine the digestibility of an insect, its ADF content may help guide its potential digestibility.

OMd showed no correlation with DH and TH and was weakly correlated with CPd. We believe that this lack of correlation may be because OMd assessed the digestibility of OM not only for protein but also for fats and carbohydrates. Therefore, OMd could not be used as a predictor of protein digestibility.

CPd is a traditional method used to measure in vitro digestibility, but it does not appear to be a good method for insects, possibly because, in its calculations, a nitrogen balance is carried out mainly using the Kjeldahl method. CPd does not distinguish between easily digestible proteins, inaccessible proteins, chitin, or other N-containing molecules [50]. DH and TH are more reliable methods because they measure the amino groups present in the sample. Furthermore, as shown in Table 3, CPd does not correlate with the other protein digestibility variables. In contrast, DH and TH show strong positive correlations with each other. In addition, the TH data reflect the maximum digestibility among the samples. The CPd data were much higher than the TH data, which could indicate an overestimation of digestibility under this method.

## 5. Conclusions

According to our results, insects can be considered a suitable protein substitute for the fishmeal and soybean meal conventionally used in feed. *T. molitor* is similar to fishmeal and soybean meal in terms of DH and TH. In addition, its meal has a better dry matter yield than that of soybean meal.

ADF showed a negative correlation with the protein digestibility variables studied. ADF content is likely related to the structures of chitin and scleroprotein in insects. Therefore, species such as *M. domestica* and *A. diaperinus* that have a high ADF content will be less digestible.

The differences found between DH and TH could indicate a possible decrease in digestibility due to ADF. More studies are needed to indisputably establish the relationship between these parameters. Finally, similar TH values demonstrated the similarity of digestibility between all the meals studied.

## Figures and Tables

**Figure 1 animals-14-00096-f001:**
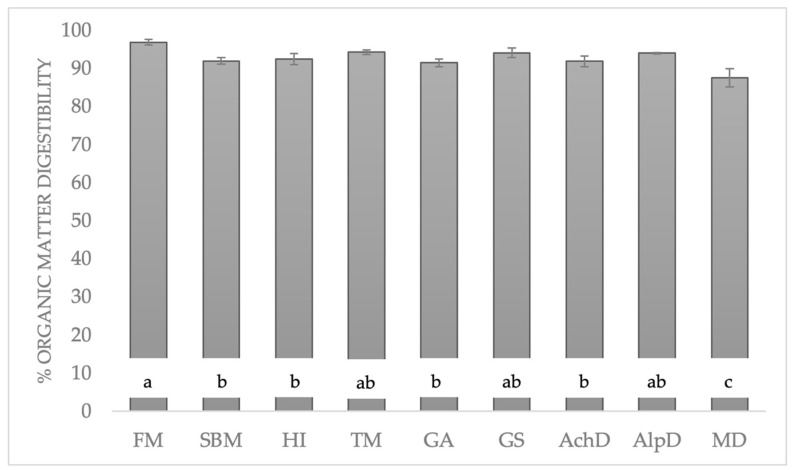
Digestibility of organic matter (%) from insects, fishmeal, and soybean meal. Significant differences are represented by different letters (*p* < 0.05). FM: fishmeal; SBM: soybean meal; HI: *H. illucens*; TM: *T. molitor*; GA: *G. assimilis*; GS: *G. sigillatus*; AchD: *A. domestica*; AlpD: *A. diaperinus*; MD: *M. domestica*.

**Figure 2 animals-14-00096-f002:**
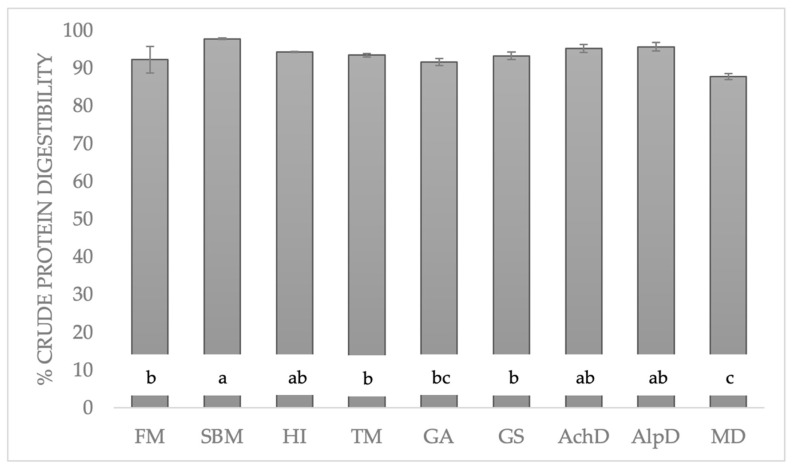
Crude protein digestibility (%) in the dry matter of insects, fishmeal, and soybean meal. Significant differences are represented by different letters (*p* < 0.05). FM: fishmeal; SBM: soybean meal; HI: *H. illucens*; TM: *T. molitor*; GA: *G. assimilis*; GS: *G. sigillatus*; AchD: *A. domestica*; AlpD: *A. diaperinus*; MD: *M. domestica*.

**Figure 3 animals-14-00096-f003:**
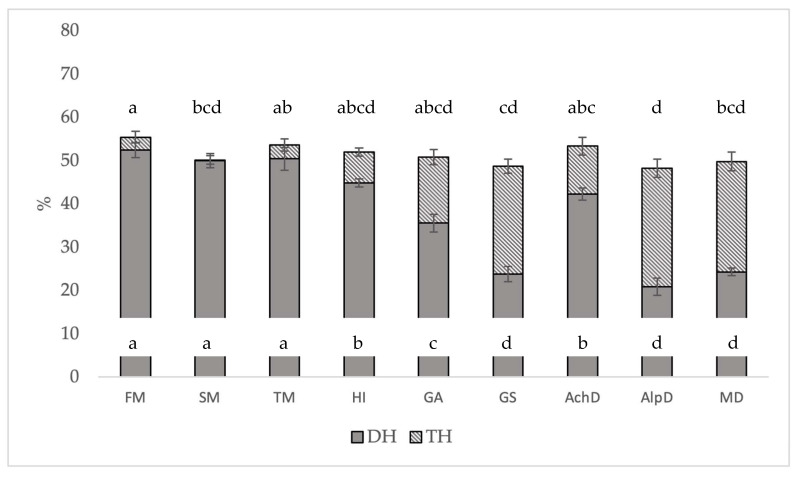
Degree of hydrolysis (DH/NH_2_) and total hydrolysis (TH/NH_2_) of insects, fishmeal, and soybean meal at the beginning and end times of protein hydrolysis, and relative standard deviation (±SD). Significant differences are represented by different letters (*p* < 0.05). FM: fishmeal; SBM: soybean meal; HI: *H. illucens*; TM: *T. molitor*; GA: *G. assimilis*; GS: *G. sigillatus*; AchD: *A. domestica*; AlpD: *A. diaperinus*; MD: *M. domestica*.

**Figure 4 animals-14-00096-f004:**
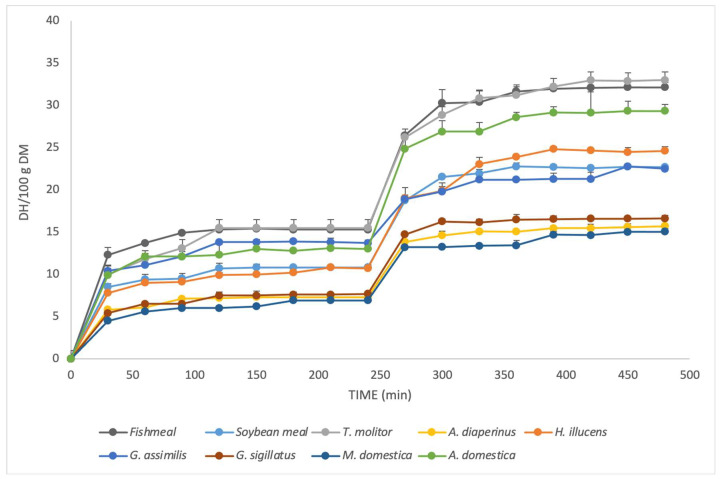
Amino groups (100 g DM) of insects, fishmeal, and soybean meal every 30 min during the gastric and intestinal phases of protein hydrolysis, and relative standard deviation (±SD).

**Table 1 animals-14-00096-t001:** Proximate chemical analysis on dry matter (%) of insects, fishmeal, and soybean meal (mean ± SD). Significant differences are represented by different letters (*p* < 0.05).

	CP ^10^	EE ^11^	ADF ^12^	Ash
FM ^1^	60.9 ± 0.6 ^bc^	14.4 ± 0.3 ^g^	7.84 ± 0.4 ^c^	19.5 ± 0.0 ^a^
SBM ^2^	50.0 ± 0.5 ^e^	1.5 ± 0.1 ^h^	8.35 ± 0.8 ^c^	6.91 ± 0.0 ^c^
HI ^3^	55.4 ± 1.3 ^d^	31.8 ± 0.4 ^a^	8.52 ± 0.5 ^bc^	8.75 ± 0.1 ^b^
TM ^4^	63.4 ± 3.2 ^b^	29.4 ± 0.3 ^b^	7.82 ± 0.3 ^c^	4.97 ± 0.0 ^f^
GA ^5^	60.4 ± 0.8 ^bc^	24.8 ± 0.4 ^c^	7.93 ± 0.3 ^c^	4.43 ± 0.0 ^g^
GS ^6^	68.9 ± 0.5 ^a^	21.7 ± 0.4 ^d^	9.20 ± 0.4 ^bc^	5.43 ± 0.0 ^d^
AchD ^7^	72.0 ± 0.3 ^a^	20.5 ± 0.4 ^e^	8.23 ± 0.3 ^c^	5.28 ± 0.1 ^e^
AlpD ^8^	72.0 ± 0.2 ^a^	18.7 ± 0.2 ^f^	9.99 ± 0.9 ^b^	4.42 ± 0.0 ^g^
MD ^9^	57.3 ± 1.2 ^cd^	29.5 ± 1.5 ^b^	16.4 ± 0.7 ^a^	5.38 ± 0.1 ^d^

^1^ fishmeal; ^2^ soybean meal; ^3^
*H. illucens*; ^4^
*T. molitor*; ^5^
*G. assimilis*; ^6^
*G. sigillatus*; ^7^
*A. domestica*; ^8^
*A. diaperinus*; ^9^
*M. domestica*; ^10^ crude protein; ^11^ ether extract; ^12^ acid detergent fiber.

**Table 2 animals-14-00096-t002:** Amino groups (100 g DM) of insects, fishmeal, and soybean meal at the start and end times of gastric (240′) and intestinal (480′) phases of protein hydrolysis, and relative standard deviation (±SD). Significant differences are represented by different letters (*p* < 0.05).

Sample	Prehydrolysis (0 min)	Gastric Phase (240 min)	Intestinal Phase (480 min)
Fishmeal	5.10 ± 0.13 ^d^	15.31 ± 0.19 ^a^	32.14 ± 0.57 ^a^
Soybean meal	3.72 ± 0.09 ^e^	10.75 ± 0.36 ^c^	22.67 ± 0.20 ^d^
*T. molitor*	13.23 ± 0.20 ^a^	15.42 ± 0.24 ^b^	32.63 ± 0.33 ^a^
*H. illucens*	10.63 ± 0.36 ^b^	10.66 ± 0.24 ^c^	24.47 ± 0.48 ^c^
*G. assimilis*	10.89 ± 0.33 ^b^	13.71 ± 0.37 ^b^	22.75 ± 0.18 ^d^
*G. sigillatus*	3.62 ± 0.17 ^e^	7.66 ± 0.17 ^d^	16.60 ± 0.38 ^e^
*A. domestica*	9.19 ± 0.76 ^b^	12.98 ± 0.89 ^b^	29.32 ± 0.77 ^b^
*A. diaperinus*	5.36 ± 0.07 ^d^	7.34 ± 0.25 ^d^	15.69 ± 0.25 ^ef^
*M. domestica*	3.31 ± 0.07 ^e^	6.87 ± 0.38 ^d^	15.03 ± 0.39 ^f^

**Table 3 animals-14-00096-t003:** Correlations between proximal composition, organic matter digestibility, crude protein digestibility, and protein digestibility (DH and total hydrolysis).

	TH	DH	OMd	CPd	ADF	CP
EE	nsd	nsd	nsd	−0.5597	nsd	nsd
CP	nsd	−0.4416 *	nsd	nsd	nsd	-
ADF	nsd	−0.5981	−0.6456	−0.5970	-	-
CPd	nsd	nsd	0.4509 *	-	-	-
OMd	nsd	nsd	-	-	-	-
DH	0.7016	-	-	-	-	-
TH	-	-	-	-	-	-

For all data, the *p*-value < 0.01, except * (*p*-value < 0.05); nsd: no significant differences.

## Data Availability

Data are contained within the article.

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
