# Peer review of "Evaluation of In Vitro Protein Hydrolysis in Seven Insects Approved by the EU for Use as a Protein Alternative in Aquaculture"

_animals, 2023, doi:10.3390/ani14010096_

Round 1
Reviewer 1 Report (Previous Reviewer 3)
Comments and Suggestions for Authors
The authors improved the manuscript. however, additional revision should be made
L15-16: delete
L19-22: rephrase "Therefore, this study evaluates the In vitro digestibility of the protein of 7 insects approved by the European Union for use in animal feed. The study suggested that Tenebrio molitor had digestibilities similar to fishmeal while Acheta domestica and Hermetia illucens obtained similar data to soybean meal."
L26: Insects, .....
There is no need for abbreviations that aren't repeated in the abstract.
L48-55: delete
L77 and others: in vivo, in vitro italic
L110: replace slaughtered with killed by freezing.
L111: at 100 C for time? Diethyl ether concentration?
L119-120: add the time of drying and combustion.
L215: delete " on dry matter"
Comments on the Quality of English Languageminor editing
Author Response
Thank you for your comments. We have corrected almost all the issues you mention.

Reviewer 2 Report (Previous Reviewer 2)
Comments and Suggestions for Authors
The manuscript “EVALUATION OF IN VITRO PROTEIN HYDROLYSIS IN 7 INSECTS APPROVED BY THE EU FOR USE AS A PROTEIN ALTERNATIVE IN AQUACULTURE” is significantly improved. However, some corrections are necessary:
Line 42: please, add “yellow” before of “mealworm”;
Line 43: replace “bed beetle” with “lesser mealworm”; replace “striped” with “banded”;
Equations (1) and (2): please, move “%” to standardize these formulas with the subsequent equations;
Equations (3), (4) e (5): please, use the same sign multiplication as the previous equations;
Equations (1)…(5): delete the numbering from all the formulas: it is already present at the end of the line;
Line 179 and 180: replace “minutes” with “min”;
Figure 3: the significance letters above the bars need to be better placed;
Line 304-305: reread the sentence and complete/delete the reference (it is an misprint?);
Line 340-341: reread the sentence (it is an misprint?).
Author Response
Thank you for your comments. We have corrected almost all the issues you mention.

Reviewer 3 Report (Previous Reviewer 1)
Comments and Suggestions for Authors
I allow myself to inform you that after reviewing and evaluating for the second time the article titled,
EVALUATION OF IN VITRO PROTEIN HYDROLYSIS IN 7 INSECTS APPROVED BY THE
EU FOR USE AS A PROTEIN ALTERNATIVE IN AQUACULTURE, (Manuscript ID: animals-2761096),
the authors have made all the suggested corrections, so the text can now be published in the journal Animals.
Author Response
Thank you for your comments.
This manuscript is a resubmission of an earlier submission. The following is a list of the peer review reports and author responses from that submission.
Round 1
Reviewer 1 Report
Comments and Suggestions for Authors
I send this information in one attachment, to the authors of the article

Reviewer 2 Report
Comments and Suggestions for Authors
Potentially, the manuscript “EVALUATION OF IN VITRO PROTEIN HYDROLYSIS IN 7 INSECTS APPROVED BY THE EU FOR USE AS PROTEIN ALTERNATIVES IN AQUACULTURE” would be interesting since it could contribute to the better knowledge of insect flour and their better use in insects-based feeds. However, the manuscript was written with little accuracy and is also insufficient in the writing style of a scientific article. Below I provide some personal opinions (general and not exhaustive) that may be of help.
The reader has little clarity of the objectives and the feeling of ambiguity of the objectives “evaluate insects and/or evaluate methods” is often present in the rest of the manuscript. Some parts (Introduction, Material and Methods, Discussion) are lacking in reference and this must also be more recent.
More details in materials and methods would be useful (e.g. "Ankon technique" [?]). The presentation of the results should be developed more and better (e.g. reporting the results of the ANOVA); some graphs are difficult to evaluate due to often illegible letters of significance. It is my opinion that the manuscript should be rewritten for resubmission.
Reviewer 3 Report
Comments and Suggestions for Authors
This in vitro study by Rodríguez-Rodríguez evaluated protein hydrolysis in 7 insects, considered a preliminary study for other animal studies. The study is good and provides data for further studies. However, the authors should revise the paper.
L17-18: unsuitable to the study. Were these proteins evaluated for human use?
L18: fishmeal
L22: the results showed that Tenebrio molitor had digestibilities ------
Again, L 24-25: Is the search for protein alternatives for humans or animals? Insect meals are protein alternatives in human diets?
L29: as feed for what?
L40-42, 45, and others: Latin names should be in italics.
The title is about protein alternatives in aquaculture, whereas a simple summary, abstract, and introduction discusses feeding insects or protein alternatives for humans. Please rewrite and focus on the aim of the study.
L105: at 100 ºC for what time? Diethyl ether, what concentration?
L112: add respectively after combustion at 500 ºC in a muffle furnace (#942.05).
L 114: acid detergent fiber
L 115: add the technique and reference.
Please complete the methods of protein hydrolysis.
L142: CPf: crude protein after hydrolysis
L 144: define OPA
L149: where are these calculations?
L150, 199, 255: degree of hydrolysis
Results: please abbreviate only the first name of the insects in the whole manuscript.
The standard abbreviation for soybean meals is SBM.
Table 1: its title should be modified—proximate chemical analysis of insects, SBM, and FM (Mean ±SD).
Please add the insects’ abbreviations in the figure legends.
L196: delete Kjeldahl method.
L200: No significant differences.
Please replace Figure 3 with a clearer one.
L232: add a reference.
The discussion needs more illustration compared with previous studies.
Comments on the Quality of English Language
The manuscript needs moderate editing.